# Plasma Inflammatory Proteome Profile in a Cohort of Patients with Recurrent Vulvovaginal Candidiasis in Kenya

**DOI:** 10.3390/jof10090638

**Published:** 2024-09-06

**Authors:** Diletta Rosati, Isis Ricaño Ponce, Gloria S. Omosa-Manyonyi, Mariolina Bruno, Nelly W. Kamau, Martin Jaeger, Vinod Kumar, Mihai G. Netea, Andre J. A. M. van der Ven, Jaap ten Oever

**Affiliations:** 1Department of Internal Medicine, Radboud Center for Infectious Diseases, Radboud University Medical Center, 6525 GA Nijmegen, The Netherlands; diletta.rosati@radboudumc.nl (D.R.); isis.ricanoponce@radboudumc.nl (I.R.P.); mariolina.bruno@radboudumc.nl (M.B.); martin.jaeger@radboudumc.nl (M.J.); v.kumar@radboudumc.nl (V.K.); mihai.netea@radboudumc.nl (M.G.N.); andre.vanderven@radboudumc.nl (A.J.A.M.v.d.V.); 2Department of Medical Microbiology & Immunology, Faculty of Health Sciences, University of Nairobi, Nairobi P.O. Box 19676, Kenya; gloria.omosa-manyonyi@radboudumc.nl; 3KAVI-Institute of Clinical Research (KAVI-ICR), Faculty of Health Sciences, University of Nairobi, Nairobi P.O. Box 19676, Kenya; wanjikukamau07@gmail.com; 4Department of Genetics, University Medical Center Groningen, University of Groningen, Hanzeplein 1, 9713 GZ Groningen, The Netherlands; 5Department of Immunology and Metabolism, Life and Medical Sciences Institute (LIMES), University of Bonn, 53115 Bonn, Germany

**Keywords:** vaginal infections, RVVC, Africa, FGF-21, proteome, inflammation

## Abstract

Vulvovaginal candidiasis (VVC) affects up to 75% of women at least once during their lifetime, and up to 8% of women suffer from frequent recurrent episodes of VVC (RVVC). A lack of a protective host response underlies vaginal *Candida* infections, while a dysregulated hyperinflammatory response may drive RVVC. This study aimed to investigate the systemic inflammatory protein profile in women with RVVC in an African population, considering the potential influence of hormonal contraceptive use on systemic inflammation. Using multiplex Proximity Extension Assay technology, we measured 92 circulatory inflammatory proteins in plasma samples from 158 RVVC patients and 92 asymptomatic women (controls). Hormonal contraceptive use was not found to have a statistically significant correlation with a systemic inflammatory protein profile in either RVVC patients or the asymptomatic women. RVVC women had lower circulating Fibroblast Growth Factor 21 (FGF-21) concentrations compared with healthy controls (adjusted *p* value = 0.028). Reduced concentrations of FGF-21 may be linked to the immune pathology observed in RVVC cases through IL-1β. This study may help to identify new biomarkers for the diagnosis and future development of novel immunomodulatory treatments for RVVC.

## 1. Introduction

Vulvovaginal candidiasis (VVC) is a clinical condition affecting up to 75% of women of childbearing age worldwide, with approximately 138 million women suffering annually from more than three episodes of acute VVC, known as recurrent VVC (RVVC) [1]. Symptoms of VVC include genital itching, redness, swelling, and pain, ultimately contributing to substantial physical and psychological stress [2]. While the pathogenesis of RVVC has not been fully elucidated, several predisposing factors have been reported to increase susceptibility to VVC/RVVC. One of the factors involved in the pathogenesis is oestrogen; this is supported by epidemiological data, which show that the incidence of VVC/RVVC rapidly increases after menarche, that pregnancy is a particularly susceptible period, and that a decrease in incidence of VVC episodes occurs after menopause [3]. Mechanistically, oestrogens promote adhesion, microbial virulence, and the immune evasion of *Candida* [3,4]. Studies on healthy women also show the systemic effects of oestrogens, as hormonal contraceptives affect the plasma concentrations of proteins involved in inflammatory and immune-related pathways [5,6].

The central feature of symptomatic VVC/RVVC is aggressive neutrophil-induced mucosal inflammation [7], which is associated with increased NLRP3 inflammasome activation and more production of pro-inflammatory cytokines, such as Interleukin (IL)-8 and IL-1β [8]. The evidence of a dysregulated systemic host response also comes from a study showing inappropriately strong cytokine production by the circulating immune cells of RVVC patients upon stimulation with *Candida* hyphae [9]. Despite these studies, a deeper understanding of the mechanisms driving non-protective *Candida* infections is urgently needed to provide women with access to new treatment options.

Studying the local dysregulated inflammatory immune response requires invasive and unpleasant investigations such as vaginal lavages or vaginal biopsies, and also poses logistical challenges. Proximity Extension Assay (PEA) technology has enabled the simultaneous targeted large-scale analysis of many inflammatory proteins in plasma, which offers possibilities to identify biomarkers of pathophysiological systemic processes that may lead to new treatment options and/or identify biomarkers of disease susceptibility [10]. To date, no published studies have investigated the circulating inflammatory protein profiles of women with RVVC. Therefore, we conducted a cohort study on Kenyan women with RVVC to determine if they display a distinct plasma proteomic signature, considering the potential influence of hormonal contraceptive use on systemic inflammatory proteins.

## 2. Methods

### 2.1. Study Population and Inclusion Criteria

This study is part of a larger prospective observational study on women with lower genital tract symptoms (LGTS) [11] and an asymptomatic comparator group, conducted between October 2018 and March 2020 at seven outpatient health facilities in Nairobi City County (NCC), Kenya. The Kenyatta National Hospital-University of Nairobi’s Ethics and Research Committee granted ethical approval for this study (P980/12/2016). A research license (permit) was obtained from the National Commission for Science Technology and Innovation, and authorization to conduct this study was obtained from NCC. Briefly, 813 symptomatic women aged between 18 and 50 years old were included after giving informed consent. The exclusion criteria were pregnancy, menopause, genital malignancy, HIV infection, and glycosuria. Social, demographic, and clinical data, including contraceptive use (hormonal and non-hormonal), were obtained from the eligible women who provided their consent. A vaginal swab was taken for microbiological examination and blood was drawn for use in proteome analysis. *Candida* infections were confirmed by direct microscopic examination and culturing on Sabouraud dextrose agar. RVVC was defined as experiencing three or more episodes of VVC in the preceding 12 months, at least one of which needed to be laboratory-confirmed (positive microscopy and/or culture). In some women with RVVC, microbiologically confirmed co-infections were present (including bacterial vaginosis [BV] and sexually transmitted infections caused by *Trichomonas vaginalis*, *Neisseria gonorrhoeae*, *Chlamydia trachomatis*, and *Mycoplasma genitalium*—for more details, see [11]). In addition, we included 104 asymptomatic women with similar exclusion criteria as the symptomatic women. From them, we also obtained EDTA plasma and performed a vaginal swab for microbiological analysis.

### 2.2. Proteomic Profiling of Circulating Inflammatory Proteins

Plasma samples were obtained from ethylenediaminetetraacetic acid (EDTA) blood by centrifugation and stored at −80 °C for proteomic analysis. Circulating plasma proteins were measured using the commercially available Proximity Extension Assay (PEA) technology from Olink^®^ Proteomics, Bioscience AB (Uppsala, Sweden) [12], using the inflammation panel (92 circulating proteins). Briefly, proteins were recognized by PEA probes, which are oligonucleotides linked to antibodies containing unique DNA sequences. Upon binding, the probes were hybridized, and the sequence was extended by a polymerase reaction. The amplified sequence was quantified by real-time PCR. The readout was expressed as log2 transformation of the Normalized Protein Expression (NPX) value and this was proportional to the protein concentration in the plasma sample. Samples were randomized and inter-plate controls were used to minimize variation. As quality control, proteins detected in less than 75% of the samples and samples that deviated by more than 0.3 NPX from the median were removed. Protein concentrations below the limit of detection were replaced by the lower limit of detection. As the samples were measured in two batches, bridge sample normalization was performed using 19 bridging samples and confirmed by principal component analyses (Appendix A). The data analysed in this manuscript are included as Appendix A.

### 2.3. Statistical Analysis

Protein abundances were assessed using a linear model included in the R limma package [13]. The linear model for contraceptive use included age as a covariate, while the model comparing RVVC women with controls included contraceptive use. Finally, the linear model used to account for the effect of co-infections included contraceptive use and co-infections as covariates. Results from these analyses were corrected for multiple testing using the Benjamini–Hochberg method, and an adjusted *p* value (adj.*p*.Val) less than 0.05 was considered statistically significant. All statistical analyses were performed in R version 4.3.2.

## 3. Results

### 3.1. Cohort Description

Drawing from the cohort of 813 women with LGTS [11] plus the 104 asymptomatic women (control), 269 women were included in this study. From these, we excluded 19 samples—18 samples failed to meet the quality control criteria and 1 sample did not have information on contraceptive use—thus leaving 250 participants for further analysis. Finally, the study population consisted of 158 (63%) symptomatic women with RVVC (cases) and 92 (37%) asymptomatic women (controls) (Figure 1). The mean age of the participants included in this study was 28 years, with a range from 19 to 50 years; for the RVCC women, the median age was 28 years (range 20–49), and for the controls it was 29 years (range 19–50). Of the total study population, 136 women (54%) used hormonal contraceptives; contraceptive use was adopted by 100 (63%) and 36 (39%) women in the cases and controls, respectively. Overall, 89 participants (36%) tested positive for at least one co-infection (BV, TV, NG, CT or MG); this was 32% and 41% of the RVVC and control groups, respectively. We did not have this information for 19 women, 16 of which were RVVC patients. *Candida* colonization was present in 8 (9%) members of the asymptomatic control group (Figure 1).

### 3.2. Women with RVVC Display the Downregulation of Circulating Fibroblast Growth Factor 21 (FGF-21) Concentration

Of the 92 plasma proteins tested, 74 (80%) were detected in at least 75% of the plasma samples and were included in the analysis. We did not observe any separation between cases and controls when using PCA that includes all the proteins (Appendix A).

We first investigated the effects of contraceptive use on the concentrations of inflammatory proteins in the entire cohort. For this, a differential abundance analysis was performed using a linear model that corrects for age. The differential abundance analysis showed that 13 proteins were significantly downregulated in women using contraceptives (Figure 2A left panel, Appendix A). Although these differences were not significant after adjustment for multiple testing (Figure 2A right panel), a clear tendency towards less inflammation was observed in women suffering from RVVC.

We then assessed the effects of contraceptive use within the RVVC group. We observed lower concentrations of four inflammatory proteins in RVVC patients using contraceptives compared to RVVC patients that did not use contraceptives, namely, TNF-related activation-induced cytokines (TRANCE), Transforming Growth Factor Alpha (TGF-alpha), TNF-related weak inducer of apoptosis (TWEAK), and Neurotrophin 3 (NT-3), and a higher concentration of Chemokine (C-C motif) ligand 28 (CCL28). While it was not statistically significant after adjustment for multiple testing (Figure 2B, right panel, Appendix A), we still included contraceptive use as a covariate in our subsequent analysis due to the observed association in unadjusted analysis.

Additionally, we compared the inflammatory profile of RVVC women with that of the controls. The differential abundance analysis showed that Fibroblast Growth Factor 21 (FGF-21) concentration (adjusted *p* value = 0.028) was significantly downregulated in RVVC patients when compared to controls (Figure 2C, Appendix A). As the women included in our cohort had various co-infections that could alter their inflammatory profile, we compared RVVC women (N = 142) with controls (N = 89), including contraceptive use and co-infection status as covariates. None of the proteins remained statistically significant after correction for multiple testing; however, FGF-21 was the most strongly downregulated protein (*p* value = 0.0007 and adjusted *p* value = 0.054, Appendix A).

## 4. Discussion

The present study is part of a larger investigation [11] aimed at characterizing the local and systemic factors contributing to susceptibility to VVC/RVVC. Here, we explored the systemic inflammatory profile in RVVC women attending seven outpatient clinics in Nairobi, Kenya. Using a targeted proteomic approach, we identified a significantly lower circulating concentration of FGF-21 in RVVC patients compared to healthy controls. To our knowledge, this is the first study to assess a broad range of circulating immunological markers in African women with RVVC.

FGF-21 is a pleiotropic hormone mainly produced by the liver and it is a regulator of glucose and lipid metabolism [14]. Higher FGF-21 plasma concentrations were found in healthy individuals from a sub-Saharan Africa population (Tanzania) compared to healthy Europeans [15]. Elevated concentrations of FGF-21 have been associated with systemic inflammatory conditions such as Crohn’s disease [16], diabetes [17], colitis [18], sepsis [19], hyperglycaemia [20], acute COVID-19 [21], and normo-uricemic gout [22]. The latter study revealed that FGF-21 attenuates IL-1β and IL-1RA production using peripheral blood mononuclear cells [22]. Extending this finding to RVVC, reduced concentrations of FGF-21 may be linked to the immune pathology observed in RVVC. This could be explained by the reduced inhibition of IL-1β production. Supportive evidence that low FGF-21 concentrations are deleterious for patients suffering from *Candida* infections comes from a study showing an upregulation of FGF-21 concentrations in patients with chronic mucocutaneous candidiasis after successful treatment with a JAK inhibitor [23].

We hypothesized that hormonal contraceptives have an immune-modifying effect, as was suggested by a recent study [6]. The absence of a statistically significant association in our study may be due to the smaller sample size of our study. Moreover, Dordevic et al. [6] examined other inflammatory proteins and also focused on the plasma proteins involved in coagulation, metabolism, and cardiovascular pathophysiology, finding the strongest associations for angiotensinogen.

Our study also has several limitations. First, plasma sampling was conducted on the day of symptomatic diagnosis and not before or in between episodes. As such, our analysis may be limited to interpretations during inflammation. To support the hypothesized causative role of FGF21 in RVVC, future studies should also sample between VVC episodes. Furthermore, the presence of co-infections may have influenced the observed inflammatory response in some women. When we adjusted for this in one of our analyses, FGF21 was no longer found to be significant after multiple testing, indicating that this may be the case. An alternative explanation for this finding is that it may be attributable to a reduction in power, given that data on co-infections were unavailable for some women. However, unlike RVVC, these infectious diseases are not typically associated with immune dysregulation. Second, we investigated the systemic protein profile without examining vaginal fluids from RVVC women, while experimental evidence suggests the importance of local dysregulation of inflammation [24]. Third, our study lacks an independent replication cohort to validate our finding. Since genetic variants may differentially affect circulatory inflammatory proteins, it would be interesting to (1) extend our study to other African populations, (2) compare our observations with the plasma proteome of RVVC women from non-African countries, and (3) investigate whether genetic variants affect any of the reported inflammatory proteins (protein quantitative trait loci, or pQTLs). This may provide a more specific signature and identify at-risk patients. Finally, the Olink platform provides relative quantification rather than absolute protein concentrations. Nevertheless, relative quantification is the predominant method used in multiplex assays in biomarker discovery due to its high specificity. The further validation of FGF-21 using commercial sandwich-based enzyme-linked immunosorbent assays (ELISAs) could be considered.

Despite these limitations, this study points out that the plasma protein FGF-21 is differentially expressed between RVVC and controls. Future functional immune measurements are warranted to validate these findings and to identify markers related to RVVC immunopathology, thereby improving diagnosis and management.

## Figures and Tables

**Figure 1 jof-10-00638-f001:**
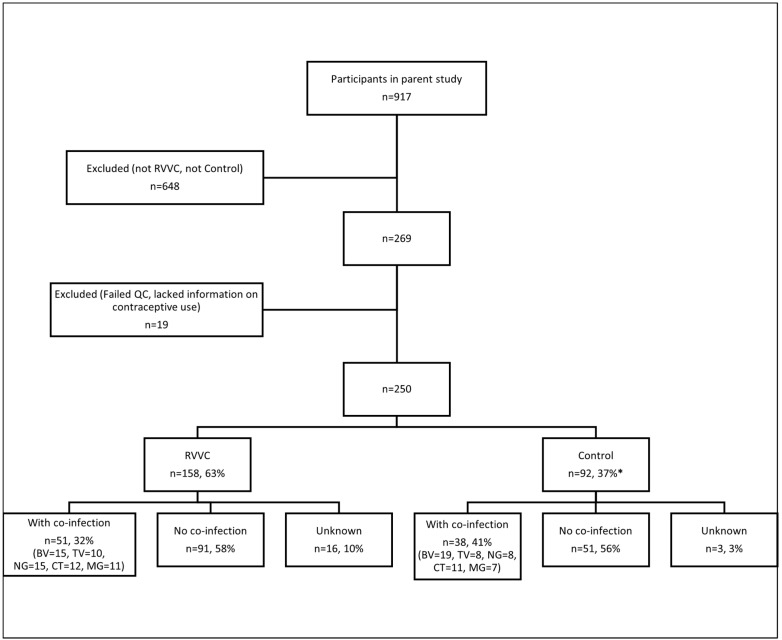
Participant inclusion flow chart. RVVC—recurrent vulvovaginal candidiasis; cases—women with RVVC; controls—asymptomatic women. BV—bacterial vaginosis; TV—Trichomonas vaginalis; NG—Neisseria gonorrhoeae, CT—Chlamydia trachomatis; MG—Mycoplasma genitalium. Key: * *Candida* colonization = 8 (8.7%).

**Figure 2 jof-10-00638-f002:**
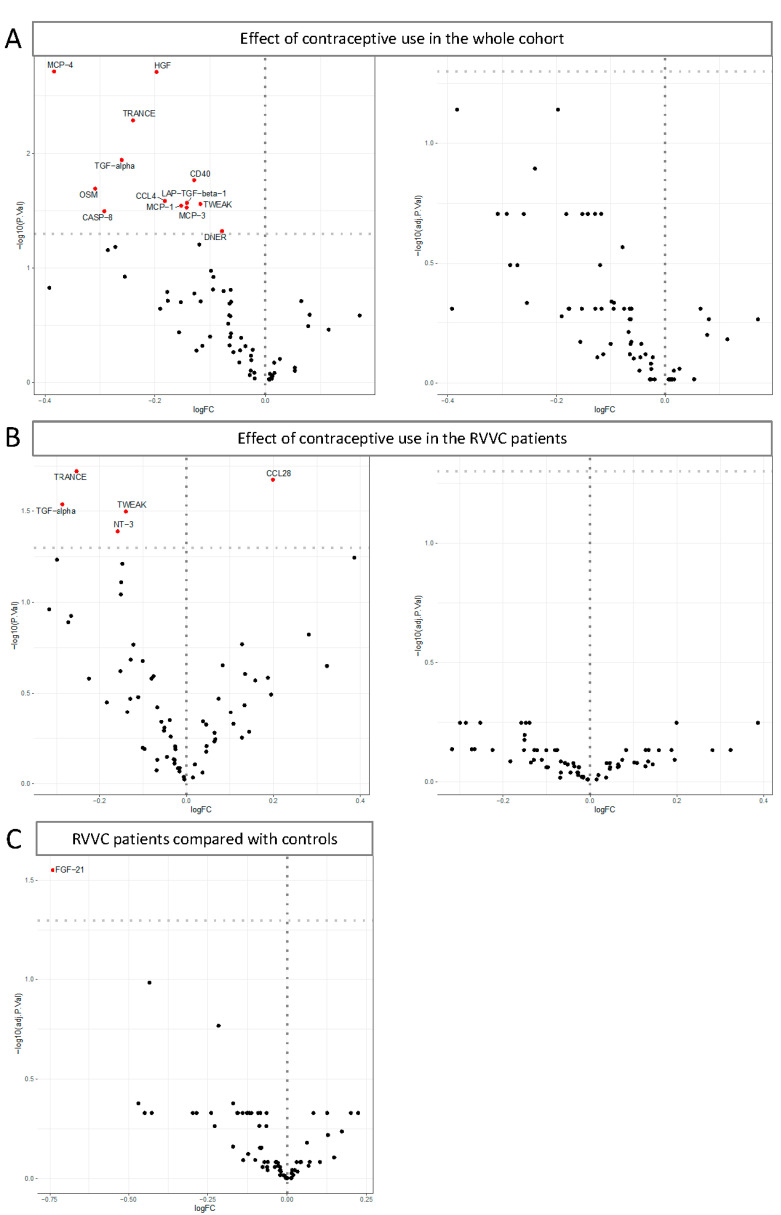
Volcano plots of 74 circulating proteins in plasma samples. (**A**) The effects of contraceptive use on inflammatory proteins concentrations in our cohort, before (**left**) and after (**right**) adjustment for multiple testing. The X axis represents the logarithm of the fold change (Log FC) of women using contraceptives compared with women that did not use contraceptives and the Y axis represents the negative logarithm base 10 of the adjusted *p* value (adj.*p*.Val). Proteins on the left are proteins that are downregulated in women using contraceptives. (**B**) The effects of contraceptive use on inflammatory proteins concentrations within the RVVC patients’ group, before (**left**) and after (**right**) adjustment for multiple testing. Proteins on the right side of the vertical dash line are higher in RVVC patients using contraceptives. The X axis represents the logarithm of the fold change (Log FC) of RVVC patients using contraceptives compared with RVVC patients that did not use contraceptives and the Y axis represents the negative logarithm base 10 of the adjusted *p* value (adj.*p*.Val). (**C**) A comparison of circulatory inflammatory proteins between RVVC patients and asymptomatic controls, adjusted for multiple testing. The X axis represents the logarithm of the fold change (Log FC) of RVVC patients compared with controls and the Y axis represents the negative logarithm base 10 of the adjusted *p* value (adj.*p*.Val) for multiple testing correction. Each protein is represented by a black dot and proteins showing a *p* value < 0.05 are depicted in red.

## Data Availability

The original contributions presented in the study are included in the article/Appendix A, further inquiries can be directed to the corresponding author.

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
