# Peer review of "Plasma Inflammatory Proteome Profile in a Cohort of Patients with Recurrent Vulvovaginal Candidiasis in Kenya"

_jof, 2024, doi:10.3390/jof10090638_

Round 1

Reviewer 1 Report

Comments and Suggestions for Authors

In this manuscript, the authors analyze plasma inflammatory proteome profile in a cohort of RVVC patients in Kenya. They found that RVVC women had lower circulating FGF-21 concentrations compared with healthy controls, and claimed that FGF-21 could be a biomarker. However, the results of this report are too preliminary and limited causing lack of innovation. Thus, I think the manuscript cannot be published in this high impact journal. The main concerns are as follows:

1. The content of the article is too thin, only containing the volcano plot result in Figure 2, and only identifying one possible biomarker. It lacks statistical comparisons and validation experiments.

2.Percent match: 24% of manuscript is too high.

3. A general comparison of samples needs to be included to illustrate the overall differences between different groups of samples.

4. KEGG and Go analyses should also be added.

5. The proteome raw data should be deposited into a public database.

Comments on the Quality of English Language

no

Author Response

Reviewer 1

In this manuscript, the authors analyze plasma inflammatory proteome profile in a cohort of RVVC patients in Kenya. They found that RVVC women had lower circulating FGF-21 concentrations compared with healthy controls, and claimed that FGF-21 could be a biomarker. However, the results of this report are too preliminary and limited causing lack of innovation. Thus, I think the manuscript cannot be published in this high impact journal. 

Answer: We thank the reviewer for taking the time to review this manuscript. Please find the detailed responses below and the corresponding revisions/corrections highlighted/in track changes in the re-submitted files.

Comments:

  1. The content of the article is too thin, only containing the volcano plot result in Figure 2, and only identifying one possible biomarker. It lacks statistical comparisons and validation experiments.

Answer: The study of recurrent vulvovaginal candidiasis is of very high importance, considering the high burden of disease. However, the vast majority of the studies on the subject have been performed in European population, and our study is the first in a population from East Africa. We acknowledge that additional validation studies in larger populations need to be performed, but due to logistical limitations this will need to be performed in future investigations.

Author’s action: We discussed this aspect as a limitation in the manuscript at Pages 7-8.

  1. Percent match: 24% of manuscript is too high.

Answer: We regret that we are unable to comprehend the reviewer's comment that "24% of the manuscript is too high." Should the reviewer be able to provide more detailed feedback, we would be better positioned to address this point; however, we are currently unable to do so.

  1. A general comparison of samples needs to be included to illustrate the overall differences between different groups of samples.

Answer: We are not sure what comparison of samples the reviewer is referring to, but we think it is a suggestion to clearly show the number of co-infections in both groups. Indeed, that information improves the clarity of the manuscript. Should the suggestion be about the proteomics analysis, the manuscript already contains all the relevant information. Olink proteomics analysis uses a relative quantification approach that measures the abundance of proteins in samples based on relative differences rather than absolute concentrations. This method utilizes Proximity Extension Assay (PEA) technology, which provides high sensitivity and specificity but does not directly measure the actual concentration of proteins. Consequently, the results are reported as Normalized Protein Expression (NPX) values, which are relative units and cannot be converted to absolute protein concentrations without external calibration standards.

To illustrate the overall differences between the groups, we have included the information about co-infections in the edited manuscript.

Author’s action: We have included in the Figure 1 the number of co-infections present in each group, as well as in the edited manuscript at Page 3 lines 127-131, as follow:

“Overall, 89 (36%) tested positive for at least one co-infection (BV, TV, NG, CT or MG); 32% and 41% of the RVVC and control groups respectively. We did not have this in-formation for 19 women, from which 16 were RVVC patients. Candida colonization was present in 8 (9%) of the asymptomatic control group (Figure 1).”

  1. KEGG and Go analyses should also be added.

Answer: Pathway enrichment analyses do provide additional insights. However, the protein panel that we used already selects proteins for their inflammatory function leading to results focused on these particular inflammatory pathways: it is not correct to perform pathway enrichment analyses, which should be performed only in case of unbiased omics analyses.

  1. The proteome raw data should be deposited into a public database.

Answer: We have now included the data that we analysed, as well as all the covariate information, as a Supplementary File 1.  We also added the following statement at Page 3, lines 105-106:

“The data analysed in this manuscript is included as Supplementary File 1” .

Reviewer 2 Report

Comments and Suggestions for Authors

In this study the authors investigated the systemic inflammatory protein profile in women with RVVC and found that the circulating concentration of FGF-21 was significantly lower in patients with RVVC compared to healthy controls.  This study represents an interesting contribution to identify circulating immunological markers for the diagnosis of RVVC and development of novel therapeutic strategies.

Following some points to improve the manuscript:

-The quality of Figure 2 should be improved

-I think the description of panel A of Figure 2 should be improved. For example, it is not clearly indicated that the 13 significantly (P value) downregulated proteins are those from RVVC women. Also: what is the comparison group?

-Please specificity if the asymptomatic women are or not colonized by Candida spp.

-How many samples failed to meet the quality control criteria? Eighteen (line 116) or nineteen (Figure 1) ?

Author Response

 Reviewer 2

In this study the authors investigated the systemic inflammatory protein profile in women with RVVC and found that the circulating concentration of FGF-21 was significantly lower in patients with RVVC compared to healthy controls. This study represents an interesting contribution to identify circulating immunological markers for the diagnosis of RVVC and development of novel therapeutic strategies.

Answer: First of all, we would like to thank the reviewer for the constructive remarks on our manuscript. We have considered and followed the comments of the reviewer and we believe that we improved the manuscript quality and readability/clarity.

Below we responded point by point to the questions raised by the reviewer.  

Comments:

1) The quality of Figure 2 should be improved.

Answer: We thank the reviewer for pointing this out. We realized the quality of the figure decreased during saving. We have now exported the figure in TIFF format, resulting in improved quality and readability.

Author’s action: Figure 2 has been replaced with a higher quality version in the revised manuscript.

2) I think the description of panel A of Figure 2 should be improved. For example, it is not clearly indicated that the 13 significantly (P value) downregulated proteins are those from RVVC women. Also: what is the comparison group?

Answer: We agree with the reviewer that the description about the groups involved in the analysis is confusing, and we apologize for that. In Figure 2 panel A we investigated the impact of contraceptive use in the entire cohort (no distinction between cases and controls), thus dividing the group in contraceptive users vs non-users. Our analysis shows that 13 significantly proteins are downregulated in women using contraceptives.

Author’s action: To improve clarity, we have indicated in the revised manuscript from which group the 13 downregulated proteins belong at Page 4 at lines 148-150 as following:

“The differential abundance analysis showed that 13 proteins were significantly down-regulated in women using contraceptives (Figure 2A left panel, Supplementary Table 1).”

We also added the following information in the figure legend at Page 7, lines 177-178:

“Proteins on the left are proteins downregulated in women using contraceptives”.

3) Please specificity if the asymptomatic women are or not colonized by Candida spp.

Author’s action: We adjusted the text and Figure 1 to improve clarity at Page 3, lines 127-131 as follows:

“Overall, 89 (36%) tested positive for at least one co-infection (BV, TV, NG, CT or MG); 32% and 41% of the RVVC and control groups respectively. We did not have this information for 19 women, from which 16 were RVVC patients. Candida colonization was present in 8 (9%) of the asymptomatic control group (Figure 1).”

4) How many samples failed to meet the quality control criteria? Eighteen (line 116) or nineteen (Figure 1) ?

Answer: We checked again all the raw data, and we confirm that the number of samples which failed the quality control criteria corresponds to 18; an additional one sample was excluded due to missing information on contraceptive use.

Author’s action:

We have corrected the revised manuscript at Page 3, lines 118-123 as follows:
“From the cohort of 813 women with LGTS [11] plus 104 asymptomatic women (control), 269 women were included in this study. From these, we excluded nineteen samples - eighteen samples failed to meet the quality control criteria and one sample did not have information on contraceptive use, thus leaving 250 participants for further analysis. Finally, the study population consisted of 158 (63%) symptomatic women with RVVC (cases) and 92 (37%) asymptomatic women (controls) (Figure 1)”.

Round 2

Reviewer 1 Report

Comments and Suggestions for Authors

The authors have made some limted revisions on their manuscript. But, I still think the the results of this report are too preliminary, and lack of crucial validation experiment. Thus, I think the manuscript cannot be pubished at current version.

Comments on the Quality of English Language

Moderate editing of English language required.

Author Response

The authors have made some limited revisions on their manuscript. But, I still think the the results of this report are too preliminary, and lack of crucial validation experiment. Thus, I think the manuscript cannot be published at current version.

Answer:  We sincerely thank the reviewer for their thoughtful feedback. We recognize the concern about the preliminary nature of our results and the lack of a validation experiments. While additional validation would indeed strengthen our findings, the current data already offers valuable insights into the pathogenesis of recurrent vulvovaginal candidiasis (RVVC) in an East African population. This manuscript fills a significant gap in the literature by providing fundamental data on RVVC in a region where such research is limited. These insights can inform future studies as suggested by the reviewer, making the current findings an important contribution to the field.

As suggested by the reviewer, we made some changes to improve the language and readability of the manuscript. The detailed changes are listed in the submitted point-by-point revision document.
